# Effects of Technostress on Psychological Contract Violation and Organizational Change Resistance

**DOI:** 10.3390/bs14090768

**Published:** 2024-09-02

**Authors:** JaeWon Shin, HyoungChul Shin

**Affiliations:** 1Divison of Business Administration, Seokyeong University, Seoul 02713, Republic of Korea; sjw740@hanmail.net; 2Department of Foodservice and Culinary Management, Kyonggi University, Seoul 03746, Republic of Korea

**Keywords:** technostress, psychological contract violation, organizational change resistance, tourism industry

## Abstract

This study investigates the effects of technostress perceived by employees on psychological contract violations and resistance to organizational change, as information and communication technology is applied to various tourism industry work sites. This study’s sample consisted of employees working in the Korean tourism industry in June 2023, who were surveyed using snowball sampling. Four hypotheses were proposed. First, technostress is associated with psychological contract violations. Second, psychological contract violations are associated with organizational change resistance. Third, technostress is associated with organizational change resistance. Fourth, psychological contract violations may play a mediating role in the relationship between technostress and organizational change resistance. All hypotheses in this study were adopted. Therefore, organizations and management in the tourism industry should identify and improve the factors that cause employee technostress arising from expanding information and communication technology to provide psychological stability.

## 1. Introduction

The tourism industry is recognized as a highly dependent industry compared to other industries, but as the method and processing speed of work are changing due to the development of information and communication technology, employees in the tourism industry are experiencing difficulties in adapting to new technologies, and various tasks such as work communication and customer consultation are also required after work, causing infringement of personal life very frequently. For this reason, it can be said that employees, who are the source of strengthening the competitiveness of tourism companies, need to prepare standards for technology using information and communication technology and manage additional tasks or stress [1]. A study by Jones et al. (2012) found that new information and communication technologies do not generate only positive aspects such as productivity and work efficiency improvement [2]. Various tourism industries have introduced information and communication technology as a strategic measure to improve corporate competitiveness and applied it to multiple fields [3], but it has been found to cause job stress for employees [4] and affect work and psychological separation, causing burnout; moreover, with the introduction of new technologies such as AI, IoT, and Big Data, which were not previously recognized at all, the older generations in the workplace have experienced maladjustment [5] in performing their work, showing extreme stress. As new information and communication technologies are applied to work, the stress that occurs due to increasing the amount of work to be processed or breaking the boundaries of work and violating personal life is called technostress [6]. According to the results of previous studies based on the transaction-based stress model, technostress affects lack of belonging, poor job performance, organizational commitment, and job dissatisfaction [6,7,8,9]. In addition, systematically, overall deviant behaviors such as absenteeism, sabotage, and waste of materials appear due to counterproductive work behavior (CWB) [10].

Like the emergence of artificial intelligence and robot technology in the hotel industry and the growth of online travel agencies, the tourism industry is actively using information and communication technology [11,12], and the resulting technostress has become a significant problem to be solved in the face of deepening technological dependence. In addition, technostress has become a research topic that must be actively dealt with regarding employee management as the reliance on technology intensifies due to the spread of intact work due to the influence of COVID-19 [13,14,15].

In other words, technostress introduces new information and communication technologies into work, and tourism workers are being asked to increase their workload and speed up their work [16]. Information and communication technology, introduced with the aim of improving corporate competitiveness, facilitates work instructions to employees even after work, infringing on daily life, and the heavy workload can cause severe stress [17]. Employees who have experienced technostress in these circumstances will recognize that the psychological contract has been violated if their obligations and resulting compensation arising from tacit consent with the organization are violated when entering the company. In addition to documented explicit agreements, there are psychological contracts based on mutual implicit expectations and promises in the employment relationship between organizations and employees, which individuals can perceive differently even in the same situation [18,19]. Various researchers have consistently argued that psychological contract violations have a very significant influence on employees’ perceptions and behaviors [20,21], and the negative consequences of this cause reactions such as job dissatisfaction and reduced organizational commitment [22]. There are no prior studies examining the direct relationship between technostress and psychological contract violations, but the relationship between technostress and psychological contract violations will be investigated based on studies showing that job stress caused by excessive job demand [23], time pressure, job complexity, and job weighting [24] affects psychological contract violations [25,26].

It can be said that, in order for a company to succeed in change, it is necessary to actively participate and cooperate with employees along with improving their constitution [27,28], and due to the expression of employees’ organizational change resistance behavior in the process of change [29], organizational change may not be carried out smoothly [30]. It was suggested that the results of organizational change resistance have a negative impact not only on employees’ negative attitudes toward their jobs and organizations but also on cooperation with colleagues and organizational productivity [27,31,32]. Considering the causes of organizational change resistance, it creates uncertainty caused by employees’ fear of change [33,34], and they want to maintain the current state and show resistance to change. In addition, dissatisfaction and distrust of the organization and management can cause resistance to organizational change [35] and problems in trust in organizations, such as psychological contract violations [19], which can lead to employee resistance in the organizational change process. Therefore, it can be seen that effective ways to reduce resistance to organizational change are significant.

As the scope of the application of new information and communication technologies to work expands, organizational changes are taking place rapidly. Failure to adapt to changes will result in the loss of competitiveness of the organization, threatening its very existence. For this reason, it is necessary to investigate the causes of resistance to low-ranking change. Therefore, despite the importance of managing technostress perceived by tourism industry employees, it can be said that studies that applied and verified the link between technostress, psychological contract violation, and organizational change resistance are insufficient. This study investigates the effect of technostress perceived by tourism industry employees on psychological contract violation and organizational change resistance. Based on these empirical studies, we would like to present academic and practical implications to tourism industry employees to adequately cope with organizational change through management measures of stress generated from information and communication technology.

## 2. Theoretical Background

### 2.1. Technostress

The word technostress is a concept created in the 1980s, when computer technology entered the business sector in earnest, and workers adapted to technological changes [8]. Technostress is defined as a state or adaptation problem caused by insufficient effort in operating a new technology, called computers as it was introduced into individuals and organizations, and as a mental disorder caused by the inability to understand new technologies [36]. Technostress is defined in various ways depending on the researcher. Still, technostress is generally defined as a “pathological response to the emergence of information technology”; it is explained that the lack of proper coping methods with new technologies causes technostress [37]. Weil and Rosen (1997) [38] stated that the adverse effects of the spread of new information technology affect users’ thoughts, attitudes, behavior, or psychological factors, and that it is a stimulating state expressed by workers who are highly dependent on information technology due to the nature of their work [5]. The stress caused by workers’ dependence on information technology, knowledge of work, and differences in users’ knowledge levels is defined as technostress [7]. Ayagari (2012) also described it as a psychological incompetence state caused by individuals who do not efficiently adapt to information technology [39].

Based on the stress theory reflecting the characteristics of information technology, research on the inducing and mitigating factors of technostress was conducted in the early stages. As research related to this progressed, studies on the consequences of technostress, technology overload, and mental effects related to role ambiguity were conducted [6,7,39,40]. Tarafdar et al. (2011) proved that problems such as techno complexity, uncertainty, widowhood, and invasion of privacy experience technostress due to personal incompetence or difficulty in adapting to rapidly developing information technology, and this stress adversely affects job performance, such as immersion, innovation, and productivity [41]. Ayyagari et al. (2011) stated that individual stress is inevitable as workers are more exposed to information and communication technology while performing their work [17]. It was also argued that the lack of adaptability to new technologies adversely affects employees’ bodies and psychology, causing job burnout [42]. Combining previous studies, it is argued that employee technostress negatively affects job satisfaction and productivity and can cause conflict by changing the work environment and destroying the balance between employees and stakeholders [43,44,45,46]. Therefore, technostress can act as a factor that makes employees feel that their psychological contract with the organization has been violated.

As a constituent factor for technostress, Brod (1984) presented four sub-factors: technology complexity, technology overload, personal life infringement, and occupational threat [36]. Tarafdar et al. (2007) [6], who conducted a study between technostress and productivity, identified five sub-factors as techno overload, techno complexity, techno invasion, techno uncertainty, and techno anxiety [7]. Brooks (2015) identified technostress as sub-factors of technostress due to techno overload, techno complexity, and techno invasion [13,47]. In this study, the direct sub-factors of technostress caused by tourism industry workers using information and communication technology were defined as techno-overload, techno-complexity, and techno-invasion. This study established the following hypothesis based on the arguments of previous studies.

Currently, there are too few studies that have directly assessed the relationship between technostress and psychological contract violations. However, a survey of role stress found that role stress directly influences psychological contract violations [48]. In particular, psychological contract violations can be predicted when a task is assigned at a given time performing multiple tasks at the same time. That is, if employees in the tourism industry are instructed using information and communication technologies that they must fulfill excessive responsibilities or tasks beyond working hours or their abilities, then they will consume additional energy and time that was not anticipated in advance and experience technological weighting. This stress causes imbalance, unfairness, and inconsistent emotions about obligations and compensation, affecting psychological contract violations. The relationship between technostress and psychological contract violations can be explained by social exchange theory [49]. According to social exchange theory, psychological contracts regarding mutual obligations begin when one of the organizations or employees fails to fulfill their commitment. Therefore, when members of an organization consume additional unexpected time and energy through technostress, they are implicitly aware of the excess of their promised work.

**Hypothesis 1.** *Technostress is associated with psychological contract violations*.

### 2.2. Psychological Contract Violation

The concept in early research on psychological contracts focused on implicit expectations with organizations or managers for employees [50] but was transformed into a study centered on mutual obligations and mutual expectations between the organization and its employees by subsequent researchers [51,52,53]. Psychological contracts are explained based on social exchange theory [54], which means that when benefits are provided by the other party, a sense of obligation to act correspondingly is formed in return, as well as recognizing the tacit commitment and approval of mutual obligations between members of the organization and the organization based on the norm of reciprocity [55]. These psychological contracts can be said to be formed mainly from two fundamental causes, and the first source arises from employees’ interrelationships with the members representing the organization. The contents of the psychological contract are recognized in the relationship with the leader or CEO of the organization where the employee works. The second source is formed by the investigator’s direct experience in organizational operation and organizational culture [56]. The implementation of appropriate psychological contracts has a positive effect on role behavior, organizational commitment, organizational citizenship behavior, and job performance of members of the organization [18,57,58,59] and weakens turnover intentions and burnout that have negative consequences for the organization [60,61].

The appropriate implementation of these psychological contracts is becoming increasingly difficult due to the rapidly changing competitive environment and the emergence and use of new technologies, and the organizational operation and organizational culture that constitute the source of psychological contracts are constantly changing. Psychological contract violations are very likely to continue in this environment [62], and psychological contract violations occur because employees perceive that the organization is not fulfilling its psychological contract obligations [63]. Psychological contract violations were defined as perceived negative emotions such as treachery by not meeting the expectations of the organization they initially perceived or negatively affecting the beliefs and expectations that form employees’ employment relationships as a result of the organization’s inaccurate implementation of the promise [64,65,66]. Robinson and Morrison (2000) divided psychological contract violations into cognitive and emotional violations. The cognitive aspect focuses on the fact that the employee has violated the psychological contract, and the emotional aspect focuses on the employee’s reaction after the psychological contract violation occurs. Therefore, the emotional aspect can be seen as linking the cognitive aspect to the employee’s attitude and behavior toward the organization [67].

This study aims to clarify the relationship between the antecedent factor of technostress that causes psychological contract violations and responses to psychological contract violations. For this purpose, the study divided the sub-factors of psychological contract violations into cognitive and emotional aspects. That is, psychological contract violations were constructed by measuring cognitive and emotional aspects and sub-factors of psychological contract violations.

Prior studies on psychological contract violations have been conducted mainly on the results of the causes and effects of psychological contract violations [21,67,68] and to identify the antecedent factors that cause psychological contract violations through various studies, negative emotions, impersonal supervision [69,70], job demand [71], employment insecurity, organizational restructuring, and injustice [72,73]. This study established the following hypothesis based on the arguments of previous studies.

Social exchange theory provides a theoretical basis for explaining the causes of psychological contract violations, which, in turn, affect organizational change resistance [49]. Moreover, social relations are strengthened based on the formation of bonds with others through exchange relationships based on mutual reciprocity. Therefore, employees posit that future exchange promises cannot be fulfilled the moment they perceive that the organization has violated a psychological contracts [74]. This transforms the relationship between the organization and its employees into an unreliable relationship [62], which engenders unfavorable attitudes and behaviors among employees [75]. Therefore, it can be concluded that tourism industry employees who witness psychological contract violations strongly resist organizational change.

**Hypothesis 2.** *Psychological contract violations are associated with organizational change resistance*.

### 2.3. Organizational Change Resistance

Organizational change can be said to be a process of trying to grow and develop by reorganizing for the survival of the organization and adjusting the internal behavior of employees according to changes in the macro and micro environment of a company. Still, employees’ reactions to this are not favorable, and resistance also appears. The moment employees perceive the organization as unstable, they reject relevant information and resist past thinking and changes [76]. Organizational change resistance is an overall negative attitude toward change, including questions about the necessity of organizational change, anxiety due to change, and elements expressed as dissatisfaction [35]. Chawla and Kelloway (2004) defined it as the “consistency of attitude or behavior that hinders organizational change goals” that resists pressure arising from organizational change [77]. Resistance means that an organization wants to maintain existing practices due to uncertainty, does not accept changes, and acts to respond to changes [78]. In situations such as mergers and acquisitions and extreme conditions such as online telecommuting, resistance to change appears more robust, and the response is expressed differently depending on individual disposition [79]. Warrick (2023) [80] suggested that the reasons for resisting organizational change can be understood in four categories: personal thinking based on employees’ low tolerance and stability in maintaining the status quo, organizational reasons, the role of change mediators, and the method of change management. In other words, resistance can be fundamentally understood as an extension of the attitude of employees to perceive change, and it can be seen that it occurs in organizational aspects due to individual characteristics and changes in the work environment and is expressed in negative attitudes such as loss and anger [81]. Although superficial change resistance, such as expressing dissatisfaction and strikes, can be adequately responded to at the organizational level, tacit resistance, such as decreased organizational loyalty, productivity, and work neglect, is challenging to manage [82]. Attempts to change organizations are causing employees to fail to fulfill existing psychological contracts they were aware of. Freese et al. (2011) [74] proved that organizational change affects employees’ performance of organizational obligations, which weakens trust in psychological contracts with the organization due to organizational change [62]. When organizational change is perceived as a threat, it appears as a form of resistance [55].

Since no prior studies have verified the direct impact relationship between technostress and organizational change resistance, we referred to a study that identified the relationship between job stress as a prerequisite. Excessive job stress arising from role conflict, role ambiguity, and role overload promotes organizational change and is closely related to employee resistance in the possible process [83]. Brod (1982) defined the phenomenon of stress caused by change as resistance to change, which leads to new stress and resistance, as a “cyclic ring”, and this ring increases the strength of resistance [84]. This study established the following hypothesis based on the arguments of several previous studies.

**Hypothesis 3.** *Technostress is associated with organizational change resistance*.

**Hypothesis 4.** *Psychological contract violations may play a mediating role in the relationship between technostress and organizational change resistance*.

This study investigated the relationship between technostress, psychological contract violation, and organizational change resistance, as shown in Figure 1.

## 3. Materials and Methods

### 3.1. Data Collection and Method

The sample of this study consisted of employees working in the Korean tourism industry (travel agencies, hotels, airlines, etc.) in 2023. Data were collected through the snow sampling approach from 1–28 June 2023. Online questionnaires were distributed through messengers to collaborators in each industry.

The ethical review and approval process, a cornerstone of our research, was meticulously conducted. It was determined that the study fell under the provisions of Article 13 of the Human Subject Research Enforcement Regulations and Article 22 of the Personal Information Protection Act, thereby waiving the need for further approval.

The collected data were analyzed using the SPSS 27.0 and AMOS 27.0 statistical analysis programs. They were analyzed using a two-step approach, and confirmatory factor analysis and reliability analysis were performed to verify the validity and reliability of the measured variables. Structural equation modeling was performed to verify the study’s hypothesis.

### 3.2. Measurement

Measurement items in this study were adopted based on previous studies, and all items were translated, modified, and evaluated on a 5-point Likert scale (“strongly disagree~strongly agree”). The scale from this study was translated into Korean by a student majoring in English literature. This process was reviewed according to the context of the Korean tourism industry by three people who hold doctoral degrees in tourism and have over ten years of experience in the tourism industry.

In this study, among the five sub-factor dimensions of technostress developed by Tarafdar et al. (2007) [6], the factor of technostress that exerts a significant influence on employees of tourism companies was measured based on previous research and as techno-overload, techno-complexity, and techno-invasion [47,85]. Based on this, it was measured by 11 questionnaires, including the item, “I think I’m being asked to do things faster due to new information technology”. To measure psychological contract violations, psychological contract violations were classified into cognitive contract violations and emotional contract violations based on the research of Robinson and Morrison (2000) [67]. Based on this, it comprised nine questionnaire items, including the item, “With new information technology, the company does not keep the contracts I made when I joined the company”. To measure crude change resistance, only behavioral resistance was set as a dependent variable based on the study of Oreg (2006) [29], and based on this, it consisted of four questionnaire items, including, “I complain to colleagues about organizational change” and “I have raised objections about organizational change”.

## 4. Results

### 4.1. Demographics of the Participants

Table 1 presents the demographic characteristics of the sample. It was distributed to 59.5% of women and 40.5% of men. In the age group, 38.9% of those in their 30s were the highest, and 59.9% of education attained was a Bachelor’s degree. Hotels accounted for the highest percentage in the type of companies respondents were working at, at 36.7%.

### 4.2. Reliability and Validity

#### 4.2.1. Confirmatory Factor Analysis

Confirmatory factor analysis results conducted to assess the feasibility of the research structure are presented in Table 2 below. Technostress and psychological contract violation are composed of secondary factors, and the sub-variables of each concept were measured as primary factors by averaging the mean points using a multiple-dimension bundle approach. Results indicate the fitness of a three-factor model (i.e., technostress, psychological contract violation, organizational change resistance) for the data based on goodness-of-fit statistics. The model fit allowed χ^2^ =80.177 (df = 24, *p* = 0.000), CMIN/DF = 3.341, RMR = 0.016, GFI = 0.968, AGFI = 0.940, NFI = 0.974, IFI = 0.982, TLI = 0.973, CFI = 0.982, and RMSEA = 0.068. In addition, the standardized factor load was 0.5 or higher, and the concept reliability was statistically significant (0.7 or higher) or higher [86]. Cronbach’s Alpha, the level of internal consistency of each structure, allowed it to be between 0.834 and 0.897 [87].

#### 4.2.2. Discriminant Validity

Table 3 shows discriminant validity. Applying the relationship between ‘technostress with the highest correlation between variables’ and ‘psychological contract violation’, the correlation coefficient between the technostress and psychological contract violation is 0.787(0.787)^2^ = 0.619. Therefore, the AVE of the technostress is 0.883, and the AVE of psychological contract violation is 0.881. Discriminant validity was found because the AVE value of the two variables was larger than the square of the correlation coefficient and the organizational change resistance AVE 0.746, which is greater than 0.619 [88].

### 4.3. Hypothesis Testing

Table 4 shows the results of SEM with AMOS 27.0 to test the hypothesis. The model goodness-of-fit is χ^2^ =80.177 (df = 24, *p* = 0.000), CMIN/DF = 3.341, RMR = 0.016, GFI = 0.968, AGFI = 0.940, NFI = 0.974, IFI = 0.982, TLI = 0.973, CFI = 0.982, RMSEA = 0.068 [86]. In the case of Hypothesis 1, the hypothesis that “Technostress is associated with psychological contract violations” was adopted (β = 0.787, *p* < 0.001). In the case of Hypothesis 2, the hypothesis that “Psychological contract violations are associated with organizational change resistance” was adopted (β = 0.431, *p* < 0.001). In the case of Hypothesis 3, the hypothesis that “Technostress is associated with organizational change resistance” was adopted (β = 0.256, *p* < 0.01). Hypothesis 4 was verified by applying the number of bootstrapping 500 samples. In the case of Hypothesis 4, the hypothesis that “Psychological contract violations may play a mediating role in the relationship between technostress and organizational change resistance” was adopted (β = 0.339, *p* < 0.01).

## 5. Discussion

Due to the development of information and communication technology, the tourism industry, which is highly dependent on human resources, has excessively overloaded employees with workload, and the adaptation process to use the advanced technology is also not easy. The new information and communication technology can cause psychological anxiety by giving tourism employees additional tasks that are different from the functions promised by the initial organization and provide an incentive to resist the flow of organizational change. Therefore, it is recognized that various efforts are required from the corporate side to reduce the technostress of members of the organization, which will be affected by the organization’s human resource management. Therefore, this study was conducted to provide theoretical and practical data for the management and development of human resources in the tourism industry by identifying the impact relationship between technostress, psychological contract violation, and organizational change resistance perceived by tourism industry employees.

The empirical analysis results of this study are as follows. Hypothesis 1 shows that the technostress experienced by employees in the tourism industry is associated with psychological contract violations. This means that employees’ efforts to adapt to information and communication technology and perform more workloads are affecting employees’ psychological contract violations. In addition, communication related to work with supervisors and customers through information and communication technology outside of the promised working hours was found to affect psychological contract violations as employees consume additional unexpected time and energy when their lives begin to be violated [67,89]. It is necessary to recognize the necessity of establishing various measures such as education to utilize new technologies to reduce the negative psychology caused by technostress by empirically verifying that information and communication technology introduced to improve work efficiency and productivity requires employees to overload or additional work, which negatively affects psychological contracts. For Hypothesis 2, it was confirmed that psychological contract violations witnessed by tourism industry employees are associated with organizational change resistance, supporting the results of previous studies [63,74,75]. As verified in Van den Heuvel and Schalk’s (2009) study, it is interpreted that the organization does not cooperate with the organization if it fails to fulfill its promises in the framework of social exchange relations based on the results that fewer people resist organizational change [90]. This shows that trust and commitment between the organization and its employees are crucial in managing human resources, one of the sources of competitiveness of tourism companies, and it is interpreted that sticking to the initially contracted work content can weaken organizational change resistance. As a result of the verification of Hypothesis 3, it was confirmed that the technostress experienced by tourism industry employees is associated with organizational change resistance; this finding supports the findings of previous studies that job stress caused by role conflict and role overload affect organizational change resistance [83,84]. This was interpreted as closely related to job stress, which also affects organizational change resistance due to information and communication technology, confirming that technostress is part of modern job stress. In light of the verification of Hypothesis 4, it can be asserted that psychological contract violations may play a mediating role in the relationship between technostress and organizational change resistance. Technostress, as described, includes factors like techno-overload, techno-complexity, and techno-invasion. These factors create stress in the workplace due to the demands and complexities of modern technology. Techno-overload refers to the pressure to work faster and longer due to technology, techno-complexity involves the challenges of managing complex technology, and techno-invasion blurs the boundaries between work and personal life by constantly connecting individuals [6,7,13,47,84]. Psychological contract violations are perceived breaches in the unwritten expectations between employees and employers [67]. These violations are divided into cognitive contract violation and emotional contract violation aspects. Cognitive contract violations might involve unmet expectations regarding job roles or responsibilities, while emotional contract violations could relate to feelings of betrayal or neglect by the employer. Organizational change resistance is the reluctance or opposition to changes within an organization. This can be influenced by various factors, including stress and perceived contract violations.

Overall, the study highlights the significant role of technostress in shaping employee attitudes and behaviors toward organizational change. By understanding these dynamics, organizations can better manage change processes and mitigate the negative impacts of technostress and psychological contract violations.

### 5.1. Theoretical Implications

The suggestions for the academic implications of this study are as follows. First, in this study, the dimension of technostress perceived by tourism industry employees due to the development of information and communication technology was composed of techno-overload, techno-complexity, and techno-invasion, and the dimension of psychological contract violation was composed of cognitive contract violation and emotional contract violation. This study is an initial effort to manage the human resources of tourism industry employees, who are more dependent on humans than other industries, by researching technostress perceived by tourism industry employees due to the development of new information and communication technology. Until now, various preceding studies have shown research by linking the preceding variables of psychological contract violation with job stress or other factors, but research on the relationship between technostress, psychological contract violation, and organizational change resistance perceived by tourism industry employees due to the development of information and communication technology is very scarce. Therefore, this study can be of academic significance by identifying the components of technostress perceived by tourism industry employees.

Various previous studies have found that job stress and psychological contract violations affect organizational change resistance, and the relationship between technostress caused by applying new information and communication technology to work and psychological contract violations and organizational change resistance perceived by employees was empirically investigated. In addition, expanding the scope of work and changing tasks beyond the promised scope of work with the organization or organization’s leaders can lead to psychological contract violations among tourism industry employees. In other words, it was confirmed that because modern society’s information and communication technology requires a workload and time that exceeds the promised scope of work for employees, employees continue to be aware of cognitive contract violations and emotional contract violations, negatively influencing the organization. According to the results of this study, it was confirmed that the technostress generated by the tourism industry’s employees using it to perform their work is increasing resistance to organizational change by frequently recognizing psychological contract violations due to the development of information and communication technology. This means that when a company’s technological business environment changes rapidly, it negatively affects the organization’s change process, increasing the likelihood of losing its competitiveness. These results are of academic significance in that they consider the importance of managing stress countermeasures for using information and communication technology in the human resource management of senior tourism industry employees. The results of this study reinforce the theoretical framework among social exchange theories. According to social exchange theory, employees can recognize that the organization is not properly overseeing their welfare and work environment when technostress occurs. This leads to the perception that the organization is not fulfilling its obligations to employees, which, in turn, can lead to psychological contract violations. In addition, employees experiencing technostress may also anticipate that the introduction of new technologies or changes to existing technologies may cause additional stress. Employees who have experienced psychological contract violations feel an imbalance in the exchange relationship with the organization. According to social exchange theory, employees recognize that fair exchange no longer exists in their relationship with the organization.

### 5.2. Practical Implications

We would like to propose the practical implications of this study as follows. First, technostress perceived by employees in the tourism industry directly affects resistance to organizational change by increasing psychological contract violations. It can be said that it is necessary to establish clear business regulations in performing tasks using new information and communication technology to prevent employees from instructing them about excessive work or after work. Technological advances have been introduced to improve work efficiency and productivity. Still, they suggest that caution is needed in using information and communication technology for work as there is a possibility that employees will continue to increase psychological contract violations and organizational change resistance. In France, a bill has been proposed and implemented to guarantee the ‘right not to be connected’ to protect employees’ personal lives after work, blocking communication channels using text messages, SNS, and phone calls. If the regulations at the corporate level are not properly followed, a bill that can protect the work label of employees as much as possible through the legislation of the National Assembly is needed in Korea. Second, psychological contract violations are affecting employees’ resistance to organizational change. As the macro and microenvironments of companies change rapidly, organizations are making various efforts to adapt to changes. However, in the process of these efforts, organizations are changing in conjunction with their relationships with employees. Since violations of psychological contracts experienced by employees in the process of change for corporate survival can be an obstacle to organizations needing change, human resource management policies that consider the aspects of employees’ psychological contracts should be established. Since success in organizational change is only possible if employees actively participate, it is necessary to fulfill their promises to employees so that the organization can be trusted. Third, psychological contract violations have a mediating effect on the relationship between technostress and organizational change resistance. Standardizing and sharing the promise of mutual obligations between the organization and its employees is essential to providing specific and accurate information on the company’s work content, workload, and welfare benefits from the employee recruitment process. In particular, technology overload and techno complexity that constitute technostress need to provide education and training to adapt to new information and communication technologies, and it is necessary to clearly define accurate work procedures and methods to prevent technology infringement that requires additional efforts without planning.

The tourism industry must respond to rapidly changing technologies and consumer demands. Therefore, the relationship between technostress, psychological contract violation, and organizational change resistance presents important implications for the tourism industry. Introducing information and communication technology (ICT) to the tourism industry is necessary; however, this can also cause technostress for workers. In addition, psychological contract violations are likely to occur because employment in the tourism industry is unstable. Moreover, continuous technology and education are required to strengthen service quality. New tourism trends and digital transformations require constant change and innovation within the industry. Therefore, the results of this study can help establish technology introduction strategies tailored to the tourism industry. This study is also anticipated to contribute to improving human resource management and establishing strategies for adapting to changes in the tourism industry.

### 5.3. Limitations and Recommendations

Future research directions based on the results of this study are as follows. Research on human resource management of tourism industry employees on technostress arising from applying new information and communication technology to work is still in its infancy, and follow-up studies are expected to be promoted. Suggestions for the limitations of this study and future research directions are as follows.

First, this study limited the research on technostress to negative reactions to new information and communication technologies by considering only stress-causing stressor factors for tourism industry employees with a very high level of human dependence compared to other industries. However, in future studies, it is necessary to apply various perspectives to identify the causal consequences of strain resulting from stress. Second, this study conducted a survey on gender, age, education level, and the type of company work in general demographics and surveyed the frequency but did not verify differences based on demographics. In future studies, if technostress perception differences are analyzed by age, education level, and type of company work, companies will be able to more effectively establish educational programs and various supports suitable for employees’ levels of new information and communication technology. Third, since this study focused on the individual level of psychological contract violations and organizational change resistance caused by the perceived technostress of tourism industry employees, there are theoretical limitations in expanding the research to the organizational level. It is believed that if research is conducted on organizational culture and organizational atmosphere from the organization’s standpoint rather than the individual level, policies that can prevent the negative effects of the organization’s technostress can be established in the future. The fourth theoretical expansion was derived by verifying the mediating effect of psychological contract violations in the relationship between technostress and organizational change resistance that previous studies of this study could not confirm. However, the mechanisms explored by simple mediation models are limited. Therefore, future studies may consider including additional variables or using more diverse methodologies to analyze the results comprehensively. Fourth, this study employed the snowball sampling method. However, the representativeness of this method may be limited because it relies heavily on the initial respondent. In addition, it is difficult to explicate causal relationships because data were collected at specific times by conducting a cross-sectional study. Therefore, it is necessary to diversify samples and employ a longitudinal study design. Fifth, because this study was conducted on Korean tourism industry workers, it may not be generalizable to other cultures and industries. That is, it is possible that the unique characteristics of Korean companies, such as their organizational culture and labor–management relations, have influenced the results. Therefore, it is necessary to analyze the variables to determine the influence of cultural factors on the results.

## 6. Conclusions

According to the results of this study, the development of innovative information and communication technology has resulted in more tasks for employees than in the past. However, the development of technology has a positive effect on the company’s performance by increasing the workload and handling tasks quickly and smoothly. However, it can be said that it is causing psychological contract violations by forcing employees to perform additional and out-of-hours tasks. Overall, technostress, psychological contract violations, and organizational change resistance perceived by employees in the process of performing their duties can provide tourism companies with a thorough education on the use of new information and communication technology, control their workload, and actively deal with the infringement of information and communication technology so that they can live their personal lives after work. Employees feel that their psychological contracts with the organization are being accurately executed, allowing them to reduce resistance in the process of organizational change.

## Figures and Tables

**Figure 1 behavsci-14-00768-f001:**
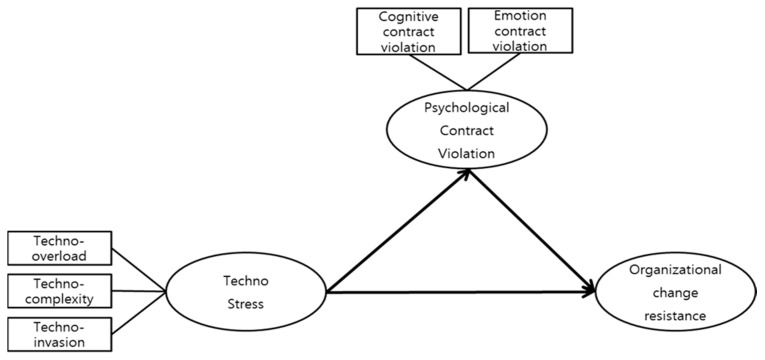
Study model.

**Table 1 behavsci-14-00768-t001:** Demographic characteristics of the participants.

Demographic Factors	Category	Number of Participants	Percentage (%)
Gender	Male	206	40.5
	Female	303	59.5
Age	20s	156	30.6
	30s	198	38.9
	40s	103	20.2
	50s and older	52	10.2
Education	High school diploma or less	97	19.1
	Associate degree	71	13.9
	Bachelor’s degree (4-year university)	305	59.9
	Graduate degree or higher	36	7.1
Type of enterprise	Travel agency	147	28.9
	Airline	142	27.9
	Hotel	187	36.7
	Etc.	33	6.5
Total		509	100

**Table 2 behavsci-14-00768-t002:** Confirmatory factor analysis.

Factor and Variable		Standardized Loading	S.E	C.R	AVE	Composite Construct Reliability(CCR)	Cronbach’s α
Technostress	Techno-overload	0.843	-	-	0.883	0.958	0.895
	Techno-complexity	0.880	0.042	23.324 ***			
	Techno-invasion	0.843	0.042	24.149 ***			
Psychological Contract Violation	Cognitive contract violation	0.861	-	-	0.881	0.937	0.834
	Emotion contract violation	0.831	0.050	19.784 ***			
Organizational Change Resistance	OCR1	0.811	-	-	0.746	0.921	0.897
	OCR2	0.824	0.044	20.819 ***			
	OCR3	0.837	0.040	21.248 ***			
	OCRI4	0.852	0.042	21.740 ***			

χ^2^ = 80.177(df = 24, *p* = 0.000), CMIN/DF = 3.341, RMR = 0.016, GFI = 0.968, AGFI = 0.940, NFI = 0.974, IFI = 0.982, TLI = 0.973, CFI = 0.982, RMSEA = 0.068. *** *p* < 0.001.

**Table 3 behavsci-14-00768-t003:** Discriminant validity.

Factor	Technostress	Psychological Contract Violation	Organizational Change Resistance
Technostress	0.883 ^(1)^	0.619 ^(3)^	0.355
Psychological Contract Violation	0.787 ** ^(2)^	0.881	0.400
Organizational Change Resistance	0.596 **	0.633 **	0.746

The notation used in the analysis is as follows: ** *p* < 0.01; ^(1)^ diagonal represents the average variance extracted (AVE); ^(2)^ the area below the diagonal represents the correlation coefficient for the constructs (r); ^(3)^ the area above the diagonal represents the square of the correlation coefficient (r^2^).

**Table 4 behavsci-14-00768-t004:** Results of structural equation model analysis.

		Estimate	t-Value	*p*-Value	Indirect Effect		Decision
	Process (Hypothesis)				Estimate	*p*	
H1	Technostress -> psychological contract violation	0.787	16.862 ***	0.000			Accepted
H2	Psychological contract violation -> organizational change resistance	0.431	5.140 ***	0.000			Accepted
H3	Technostress -> organizational change resistance	0.256	3.204 **	0.001			Accepted
H4	Inclusive leadership -> organizational change resistance (mediating effects of psychological contract violation)				0.339 **	0.004	Accepted

** *p* < 0.01; *** *p* < 0.001.

## Data Availability

The data presented in this study are available on request to the corresponding author.

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
