# Peer review of "Effects of Technostress on Psychological Contract Violation and Organizational Change Resistance"

_behavsci, 2024, doi:10.3390/bs14090768_

Round 1

Reviewer 1 Report

Comments and Suggestions for Authors

The article provides a comprehensive examination of the relationships between technostress, psychological contract violations, and organizational change resistance among employees in the tourism industry. By focusing on these interconnected factors, the study sheds light on the challenges faced by employees in technology-driven work environments and the implications for organizational dynamics.

One notable strength of the article is its clear articulation of the research objectives and the theoretical framework guiding the study. The authors effectively define key concepts, such as technostress and psychological contract violations, and establish a solid foundation for investigating their impact on organizational change resistance. The empirical analysis conducted in the study adds valuable insights to the existing literature on these topics, contributing to a deeper understanding of the complexities involved in managing technostress in the workplace.

Moreover, the article's discussion of the practical implications for organizations in the tourism industry highlights the relevance of the research findings for human resource management and organizational development. By offering recommendations for mitigating technostress and addressing psychological contract violations, the study provides actionable insights that can inform strategies for enhancing employee well-being and organizational effectiveness.

I leave here some recommendations that can strengthen the article:

The introduction could benefit from a more detailed discussion on why the tourism industry was chosen as the focus of the research. Providing specific reasons or industry-related challenges that make this sector particularly susceptible to technostress could enhance the justification for the study's focus.

- Additionally, the introduction could benefit from a more comprehensive review of existing literature on technostress, psychological contract violations, and organizational change resistance. While some references are provided, a more in-depth discussion of prior research findings and theoretical frameworks related to these concepts could strengthen the theoretical foundation of the study.

The theoretical background section of the study provides a detailed overview of the concept of technostress and its relevance to the tourism industry. The section effectively defines technostress and highlights its various dimensions, such as techno overload, techno complexity, and techno invasion. By incorporating definitions from multiple researchers, the section offers a comprehensive understanding of technostress and its implications for individuals working in technology-dependent environments.

However:

- it could benefit from a more critical analysis of the existing literature on the topic. Including discussions on conflicting perspectives or unresolved debates in the field of technostress could add depth to the theoretical framework.

- it could expand on the theoretical underpinnings of psychological contract theory and its implications for organizational behavior. By delving deeper into the theoretical frameworks that underpin psychological contract violations and organizational change resistance, the theoretical background could enhance the reader's understanding of the conceptual framework guiding the study.

The results section includes a structured presentation of participant demographics, aiding in understanding the sample profile and potential generalizability. Additionally, the section effectively communicates the outcomes of hypotheses testing, providing transparency in reporting the research findings and their implications.

The discussion section effectively connects the results to the existing literature and theoretical frameworks, offering insights into the implications of the study for both theory and practice. However it could expand on the practical implications of the study for organizations in the tourism industry. Providing specific recommendations for managing technostress, addressing psychological contract violations, and reducing resistance to organizational change could offer valuable guidance for practitioners and human resource professionals in the industry.

Author Response

Dear reviewer

We express our sincere gratitude for your invaluable guidance on our manuscript. Your comments are crucial to us and have significantly contributed to enhancing the quality of our work. Following your feedback, we have revised the initial manuscript further, with all changes highlighted in red. While the revised manuscript may still have some imperfections, we kindly ask for your ongoing suggestions to further improve it. Your continuous guidance is crucial for meeting the publication standards set by the behavioral sciences journal.  Once again, we thank you for your hard work and patient guidance. Below are the details of the amendments:

Reviewer

Comment 1: The article provides a comprehensive examination of the relationships between technostress, psychological contract violations, and organizational change resistance among employees in the tourism industry. By focusing on these interconnected factors, the study sheds light on the challenges faced by employees in technology-driven work environments and the implications for organizational dynamics.

One notable strength of the article is its clear articulation of the research objectives and the theoretical framework guiding the study. The authors effectively define key concepts, such as technostress and psychological contract violations, and establish a solid foundation for investigating their impact on organizational change resistance. The empirical analysis conducted in the study adds valuable insights to the existing literature on these topics, contributing to a deeper understanding of the complexities involved in managing technostress in the workplace.

Moreover, the article's discussion of the practical implications for organizations in the tourism industry highlights the relevance of the research findings for human resource management and organizational development. By offering recommendations for mitigating technostress and addressing psychological contract violations, the study provides actionable insights that can inform strategies for enhancing employee well-being and organizational effectiveness.

I leave here some recommendations that can strengthen the article:

The introduction could benefit from a more detailed discussion on why the tourism industry was chosen as the focus of the research. Providing specific reasons or industry-related challenges that make this sector particularly susceptible to technostress could enhance the justification for the study's focus.

- Additionally, the introduction could benefit from a more comprehensive review of existing literature on technostress, psychological contract violations, and organizational change resistance. While some references are provided, a more in-depth discussion of prior research findings and theoretical frameworks related to these concepts could strengthen the theoretical foundation of the study.

The theoretical background section of the study provides a detailed overview of the concept of technostress and its relevance to the tourism industry. The section effectively defines technostress and highlights its various dimensions, such as techno overload, techno complexity, and techno invasion. By incorporating definitions from multiple researchers, the section offers a comprehensive understanding of technostress and its implications for individuals working in technology-dependent environments.

However:

- it could benefit from a more critical analysis of the existing literature on the topic. Including discussions on conflicting perspectives or unresolved debates in the field of technostress could add depth to the theoretical framework.

- it could expand on the theoretical underpinnings of psychological contract theory and its implications for organizational behavior. By delving deeper into the theoretical frameworks that underpin psychological contract violations and organizational change resistance, the theoretical background could enhance the reader's understanding of the conceptual framework guiding the study.

The results section includes a structured presentation of participant demographics, aiding in understanding the sample profile and potential generalizability. Additionally, the section effectively communicates the outcomes of hypotheses testing, providing transparency in reporting the research findings and their implications.

The discussion section effectively connects the results to the existing literature and theoretical frameworks, offering insights into the implications of the study for both theory and practice. However it could expand on the practical implications of the study for organizations in the tourism industry. Providing specific recommendations for managing technostress, addressing psychological contract violations, and reducing resistance to organizational change could offer valuable guidance for practitioners and human resource professionals in the industry.

Response 1: Dear reviewer, We appreciate your guidance. We have revised the manuscript based on your comments. Your review is kindly requested.

We have added the following sentences to the introduction. "Like the emergence of artificial intelligence and robot technology in the hotel industry and the growth of online travel agencies, the tourism industry is actively using information and communication technology (Srinivasan et al., 2002; Lin & Hsieh, 2011), and the resulting technostress has become a significant problem to be solved in the face of deepening technological dependence. In addition, technostress has become a research topic that must be actively dealt with regarding employee management as the reliance on technology intensifies due to the spread of intact work due to the influence of COVID-19. (Molino et al., 2020; Nimrod, 2022; Camacho & Barrios, 2022)."

We have restated the description of the sub-factors that make up the psychological breach of contract. The contents are as follows.

Robinson and Morrison (2000) divided psychological contract violations into cognitive and emotional violations. The cognitive aspect focuses on the fact that the employee has violated the psychological contract, and the emotional aspect focuses on the employee's reaction after the psychological contract violation occurs. Therefore, the emotional aspect can be seen as linking the cognitive aspect to the employee's attitude and behavior toward the organization.

In other words, psychological contract violations were constructed by measuring the cognitive and emotional aspects, sub-factors of psychological contract violations.

Thank you for your valuable opinion. We constructed the conclusions based on your advice and added the limitations of this study. For the revised content, we added the following sentences.

  1. Conclusions

According to the results of this study, the development of innovative information and communication technology has resulted in more tasks for employees than in the past. However, the development of technology has a positive effect on the company's performance by increasing the workload and handling tasks quickly and smoothly. However, it can be said that it is causing psychological contract violations by forcing employees to perform additional and out-of-hours tasks. Overall, technostress, psychological contract violations, and organizational change resistance perceived by employees in the process of performing their duties can provide tourism companies with a thorough education on the use of new information and communication technology, control their workload, and actively deal with the infringement of information and communication technology so that they can live their personal lives after work. Employees feel that their psychological contracts with the organization are being accurately executed, allowing them to reduce resistance in the process of organizational change.

The fourth theoretical expansion was derived by verifying the mediating effect of psychological contract violations in the relationship between technostress and organizational change resistance that previous studies of this study could not confirm. However, the mechanisms explored by simple mediation models are limited. Therefore, future studies may consider including additional variables or using more diverse methodologies to analyze the results comprehensively.

Reviewer 2 Report

Comments and Suggestions for Authors

In this manuscript, the authors investigate the effects of technostress perceived by employees on psychological contract violations and resistance to organizational change. Utilizing snowball sampling, they conducted a survey among Korean tourism employees. The results indicate that psychological contract violations mediate the relationship between technostress and resistance to organizational change. However, several concerns need to be revised:

1.The hypotheses require more compelling evidence. For instance, on page 3, the relationship between technostress and psychological contract violations is not clearly articulated from the preceding argument. It is recommended to reorganize the logical structure of the introduction section.

2.On page 4, line 1, hypothesis 1, the sentence may cause confusion for readers. It would be beneficial if the authors could clearly delineate the two main factors.

3.The mechanisms explored by a simple mediation model are limited. The authors might consider incorporating additional variables or employing more diverse methodologies to provide a more comprehensive analysis of the results.

Comments on the Quality of English Language

The quality of English Language in this manuscript is fine. However, some minor editing of English language is required. After some polish, it could be better.

Author Response

Dear reviewer

We express our sincere gratitude for your invaluable guidance on our manuscript. Your comments are crucial to us and have significantly contributed to enhancing the quality of our work. Following your feedback, we have revised the initial manuscript further, with all changes highlighted in red. While the revised manuscript may still have some imperfections, we kindly ask for your ongoing suggestions to further improve it. Your continuous guidance is crucial for meeting the publication standards set by the behavioral sciences journal.  Once again, we thank you for your hard work and patient guidance. Below are the details of the amendments:

Reviewer

Comment 1: 1.The hypotheses require more compelling evidence. For instance, on page 3, the relationship between technostress and psychological contract violations is not clearly articulated from the preceding argument. It is recommended to reorganize the logical structure of the introduction section.

Response 1: Dear reviewer, We appreciate your guidance. We have revised the manuscript based on your comments. Your review is kindly requested.

We have added the following sentences to the introduction. "Like the emergence of artificial intelligence and robot technology in the hotel industry and the growth of online travel agencies, the tourism industry is actively using information and communication technology (Srinivasan et al., 2002; Lin & Hsieh, 2011), and the resulting technostress has become a significant problem to be solved in the face of deepening technological dependence. In addition, technostress has become a research topic that must be actively dealt with regarding employee management as the reliance on technology intensifies due to the spread of intact work due to the influence of COVID-19. (Molino et al., 2020; Nimrod, 2022; Camacho & Barrios, 2022)."

In addition, we added the following statement to enhance the logical basis of Hypothesis 1, "Therefore, technostress can act as a factor that makes employees feel that has been violated."

Comment 2: On page 4, line 1, hypothesis 1, the sentence may cause confusion for readers. It would be beneficial if the authors could clearly delineate the two main factors.

Response 2: Dear reviewer, We appreciate your guidance. We have revised the manuscript based on your comments. Your review is kindly requested.

We have rewritten the description of the components of a psychological breach of contract by your noble opinion. The contents are as follows.

“Robinson and Morrison (2000) divided psychological contract violations into cognitive and emotional violations. The cognitive aspect focuses on the fact that the employee has violated the psychological contract, and the emotional aspect focuses on the employee's reaction after the psychological contract violation occurs. Therefore, the emotional aspect can be seen as linking the cognitive aspect to the employee's attitude and behavior toward the organization.”

In addition, we have added the following sentences to improve readers' understanding according to the noble opinions of our reviewer.

“In other words, psychological contract violations were constructed by measuring the cognitive and emotional aspects, sub-factors of psychological contract violations.”

Comment 3: The mechanisms explored by a simple mediation model are limited. The authors might consider incorporating additional variables or employing more diverse methodologies to provide a more comprehensive analysis of the results.

Response 3: Dear reviewer, We appreciate your guidance. We have revised the manuscript based on your comments. Your review is kindly requested.

We greatly appreciate your sharp point. Although this study was the first to attempt to verify the mediating effect of psychological contract violations in the relationship between technostress and organizational change resistance, we agree that it has limitations, as you said. Therefore, we wanted to include your opinions in the section on the limitations of this study and future research directions. The contents are as follows.

The fourth theoretical expansion was derived by verifying the mediating effect of psychological contract violations in the relationship between technostress and organizational change resistance that previous studies of this study could not confirm. However, the mechanisms explored by simple mediation models are limited. Therefore, future studies may consider including additional variables or using more diverse methodologies to analyze the results comprehensively.

Round 2

Reviewer 2 Report

Comments and Suggestions for Authors

The authors have responded to the comments and revised the manuscript with evidence. However, there are still some concerns:

1. The hypotheses still require more compelling evidence. For example, on page 4, line 180, you have added some sentences to strengthen the logic of hypothesis 1. However, the sentences you added show the importance of technostress, not the relationship between technostress and psychological contract violations. It is recommended that the logical structure of the introduction section be reorganized.

2. The result needs further discussion. In your study, the independent variables and the mediator variables have multiple dimensions. However, in the discussion part, there is little explanation about the effects of the different dimensions. It is suggested to add further discussion to enrich your study.

Comments on the Quality of English Language

Minor editing of English language required.

Author Response

Dear reviewer

We express our sincere gratitude for your invaluable guidance on our manuscript. Your comments are crucial to us and have significantly contributed to enhancing the quality of our work. Following your feedback, we have revised the initial manuscript further, with all changes highlighted in red. While the revised manuscript may still have some imperfections, we kindly ask for your ongoing suggestions to further improve it. Your continuous guidance is crucial for meeting the publication standards set by the behavioral sciences journal. Once again, we thank you for your hard work and patient guidance. Below are the details of the amendments:

Reviewer

Comment 1:  The hypotheses still require more compelling evidence. For example, on page 4, line 180, you have added some sentences to strengthen the logic of hypothesis 1. However, the sentences you added show the importance of technostress, not the relationship between technostress and psychological contract violations. It is recommended that the logical structure of the introduction section be reorganized.

Response 1: Dear reviewer, We appreciate your guidance. We have revised the manuscript based on your comments. Your review is kindly requested. The study of technostress on psychological contract violation and organizational change resistance is a study that explored the causal relationship between variables that did not exist before. Based on your sincere advice, we have added the following sentences.

“No studies have yet explored the direct effect of technostress on psychological contract violations. Nevertheless, it can be inferred that technostress affects psychological contract violations based on several previous studies.”

Comment 2: The result needs further discussion. In your study, the independent variables and the mediator variables have multiple dimensions. However, in the discussion part, there is little explanation about the effects of the different dimensions. It is suggested to add further discussion to enrich your study.

Response 2: Dear reviewer, We appreciate your guidance. We have revised the manuscript based on your comments. Your review is kindly requested. Based on your sincere advice, we have added the following sentences.

 “Technostress, as described, includes factors like techno-overload, techno-complexity, and techno-invasion. These factors create stress in the workplace due to the demands and complexities of modern technology. Techno-overload refers to the pressure to work faster and longer due to technology, techno-complexity involves the challenges of managing complex technology, and techno-invasion blurs the boundaries between work and personal life by constantly connecting individuals [5, 6, 12, 35, 46]. Psychological contract violations are perceived breaches in the unwritten expectations between employees and employers [65]. These violations are divided into cognitive contract violation violation and emotional contract violation aspects. Cognitive contract violations might involve unmet expectations regarding job roles or responsibilities, while emotional contract violations could relate to feelings of betrayal or neglect by the employer. Organizational change resistance is the reluctance or opposition to changes within an organization. This can be influenced by various factors, including stress and perceived contract violations.

Overall, the study highlights the significant role of technostress in shaping employee attitudes and behaviors toward organizational change. By understanding these dynamics, organizations can better manage change processes and mitigate the negative impacts of technostress and psychological contract violations.”
